**communications** engineering

# Submersible touchless interactivity in conformable textiles enabled by highly selective overbraided magnetoresistive sensors

Pasindu Lugoda ®[1,5] ✉, Eduardo Sergio Oliveros-Mata[2,5], Kalana Marasinghe[3], Rahul Bhaumik[4], Niccolò Pretto ®[4], Carlos Oliveira[3], Tilak Dias[3], Theodore Hughes-Riley ®[3] ✉, Michael Haller ®[4], Niko Münzenrieder ®[4] ✉ & Denys Makarov ®[2] ✉

Miniature electronics positioned within textile braids leverages the persistent flexibility and comfort of textiles constructed from electronics with 1D form factors. Here, we developed touchless interactivity within textiles using 1D overbraided magnetic field sensors. Our integration strategy minimally impacts the performance of flexible giant magnetoresistive sensors, yielding machine-washable sensors that maintain conformability when integrated in traditional fabrics. These overbraided magnetoresistive sensors exhibit a detectivity down to 380 nT and a nearly isotropic magnetoresistance amplitude response, facilitating intuitive touchless interaction. The interactivity is possible even in humid environments, including underwater, opening reliable activation in day-to-day and specialized applications. To showcase capabilities of overbraided magnetoresistive sensors, we demonstrate a functional armband for navigation control in virtual reality environments and a self-monitoring safety helmet strap. This approach bridges the integration gap between on-skin and rigid magnetic interfaces, paving the way for highly reliable, comfortable, interactive textiles across entertainment, safety, and sportswear.

Electronic textiles are becoming increasingly popular and have seen use in applications ranging from personalised healthcare to flexible displays[1–5]. The reasons for this are manyfold: Fabrics are typically shapeable, pleasant to look at, and comfortable to feel. For wearable applications in particular, clothes are in a persistent intimate contact with the human body which makes them an ideal platform to collect physiological data, as well as to act as an unobtrusive interface between the physical reality of humans and the virtual domain. This becomes progressively more important for entertainment and professional applications. The fusion of electronic functionality and textile fabrics present challenges due to the different mechanical properties of textiles and traditional electronics[6]. Textiles have benefited by the evolution of the form factor of electronics that have evolved from bulky three-dimensional (3D) devices through 2D film based flexible electronics all the way to 1D fibre-like formats. 1D form factors increase the conformability of constructed textiles by leveraging the increased flexibility in the plane perpendicular to the fibre[7]. Integrating electronic components in a fibre stimulated a new research and technology field[8,9].

There is an active research on the realization of unobtrusive smart textiles relying on soft and flexible materials[1,10]. Some of the explored approaches include constructing sensors from conductive yarns[11], attaching miniaturised rigid devices on textile substrates[12], printing components on textile substrates[13], weaving flexible electronics into fabrics[14], or by coating and functionalizing individual fibres[15,16]. While these strategies may enable discreet integration of electronics into textiles, most are not mechanically robust enough to endure washing, bending, and shearing of the textile without compromising the device's functionality.

Due to the unobtrusive nature of electronic textiles, they have been recent efforts to use them as interactive controls for robots[17], touch displays[2],

[1]Department of Engineering, School of Science and Technology, Nottingham Trent University, Nottingham, UK. [2]Helmholtz-Zentrum Dresden-Rossendorf e.V., Institute of Ion Beam Physics and Materials Research, Dresden, Germany. [3]Advanced Textiles Research Group, Nottingham School of Art and Design, Nottingham Trent University, Nottingham, UK. [4]Faculty of Engineering, Free University of Bozen-Bolzano, Bozen-Bolzano, Italy. [5]These authors contributed equally: Pasindu Lugoda, Eduardo Sergio Oliveros-Mata. ✉e-mail: pasindu.lugoda@ntu.ac.uk; theo.hughes-riley@ntu.ac.uk; niko.muenzenrieder@unibz.it; d.makarov@hzdr.de

and augmented/virtual reality[18]. A variety of electronic textiles has been developed for monitoring temperature[19,20], strain[21–23], pressure[20,24], moisture/humidity[25,26], or acceleration[27]. There are already established smart textile technologies for illumination[28], energy harvesting[29,30], and sensing physiological parameters[1,31]. Currently the domain of textile based interactive surfaces is dominated by *tactile* sensors capturing mechanical stimuli[5,32]. The advantage of tactile sensors is that these can be manufactured using diverse transduction mechanisms such as resistive/piezoresistive[33–37], capacitive[38–43], triboelectric[33,44–46], optical[47], and piezoelectric approaches[48,49]. This variety of approaches in turn opens the possibility to utilize many different materials and process, some of which are compatible with textiles, for the fabrication of such interfaces. There is, however, a disadvantage associate to this approach, namely that these technologies are prone to accidental activation when integrated into wearable systems for conformable textiles. Accidental activation may occur when the sensor comes into contact or rubs against another surface. This limits their usefulness when used in textiles which constantly rub or brush against different surfaces when used in everyday scenarios. Therefore, there has been a recent trend towards touchless interactivity in textiles.

Touchless interaction effectively reduces the wear and tear, has gesture recognition capabilities and allows for increased aesthetic control of functional textiles. Table 1 showcases the latest technologies utilized in this area. Capacitive sensing technology, capable of capturing electric fields and functioning as proximity sensors, emerges as one of the most utilized technologies in creating touchless interactive textiles[40–42,50]. Many of the technologies listed in Table 1, including capacitive ones, can be unintentionally activated by nearby objects. This makes them less suitable for smart textile applications, where textiles come into close proximity with multiple objects during daily wear. Meanwhile, other touchless interactive technologies require users to remember specific gestures to interact with the textile[51]. This also entails extensive post-processing of sensor signals to ensure activation through these gestures. Integrating touchless interactivity into textiles presents unique challenges towards real applications. The design must account for the demanding mechanical properties, constant deformation, washing requirements, and operation in humid conditions inherent to textile materials.

Maintaining the mechanical behaviour of a textile is far more achievable when electronics take on a 1D geometry, such as a fibres or yarns, rather than the classical 2D patches[7]. These formats are compatible with the interweaving of the textile, resulting in excellent flexibility in the plane perpendicular to the fibre or yarn. However, this ease of deformation often leads to unstable readout during deformation making the system unreliable during bending and stretching. Furthermore, the need for washing textiles increases the risk of failure for electronics that cannot be fully encapsulated. Finally, interactive wearable textiles are frequently exposed to humidity from rain, sweat, and even underwater scenarios, posing challenges for classical touchless approaches based on capacitive activation. No single platform currently satisfies all these requirements in a 1D form factor for smart textiles.

Utilizing magnetic fields for the activation of sensors may provide a more suitable approach to enable intuitive, reliable, and selective textile integrated interfaces[52]. Magnetic flux densities that are significantly larger than the geomagnetic field (25–65 μT)[53] can easily be created by miniaturized permanent magnets and be used to trigger magnetic sensors. This allows for the creation of more selective interfaces when compared to conventional approaches. Furthermore, the detection of such fields does not require direct physical contact[54,55]. Magnetic sensors integrated into textiles allow to quantify the absolute position, orientation, and the relative movement in a static or dynamic magnetic field[52,56–58]. The challenge towards the realization of these prospects is to fabricate and integrate magnetic field sensors that are mechanically compatible with textiles and with sufficient sensing performance. Hence, there is only limited work around the combination of magnetic sensors and textiles. Supplementary Table S1 presents a comparison of our magnetic sensing technology against other available magnetic sensing technologies in textiles. In fact, washable

**Table 1 | Comparison of various smart textiles with touchless interactivity functions**

| Ref | Structure | Sensing technology | Targeted activation | Flexible sensing element | Smart yarn/fibre | Deformability | Machine washable | Functional underwater | Application |
|---|---|---|---|---|---|---|---|---|---|
| 41 | PDMS encapsulated rigid electronic on polyimide strips | Capacitive | No | No | No | Bending, twisting | Yes | N/A | Proximity, touch detection |
| 72 | BC–BC/Gr helical fibres | Capacitive | No | Yes | Yes | Folding, bending | N/A | N/A | HMI, motion tracking |
| 51 | Conductive thread embroidered onto a fabric | Doppler | No* | Yes | No | N/A | N/A | N/A | Gesture recognition |
| 73 | Piezoelectric material and electrodes sandwiched in between two magnets | Ultrasound | No* | No | No | N/A | N/A | N/A | Gesture recognition |
| 74 | Semiconductor attached to tungsten wires and embedded in a fibre cladding | Photosensitive | Yes | No | Yes | N/A | N/A | N/A | Communication, photoplethysmography |
| 37 | Spandex fabric coated with $Fe_3O_4$ nanoparticles | Magnetic | Yes | Yes | No | Stretching, bending | Yes | N/A | HMI, motion tracking |
| This work | Flexible GMR sensor embedded within a braided structure | Magnetic | Yes | Yes | Yes | Bending, shearing, twisting and draping | Yes | Yes | Underwater HMI, safety wear, VR |

"*" indicates that gesture recognition was enabled by software.

and mechanically conformal 1D magnetic sensors have never been unobtrusively integrated within textiles and used for human computer interactions. In the past, rigid magnetic field sensors have been applied for motion tracking systems of smart gloves by companies such as HaptX[59]. However, attaching such solid devices to the textiles surface makes the textile rigid and uncomfortable to wear. Additionally, proximity detection was permed by a spandex fabric coated with $Fe_3O_4$ nanoparticles[37]. Such textile coatings change drastically the mechanical properties of the textile and are generally impacted by hysteresis, low speed and are typically not resistant to abrasion[60].

Here, we demonstrate seamless integration of the reliable touchless interactivity into everyday and specialized clothing using flexible, high-performance magnetoresistive sensors. Nanostructured Cu/Co giant magnetoresistive (GMR) sensors fabricated on flexible substrates were integrated within a braided textile yarns to ensure compatibility with conventional textile manufacturing equipment, continuous and reliable electrical connections, as well as to encapsulate the devices, and to preserve the feel and the aesthetics of the textile. These braided textile structures will henceforth be referred to as overbraided magnetoresistive sensors. These overbraided magnetoresistive sensors exhibit exceptional sensing performance with sub-µT detectivity, can be cleaned in a conventional washing machine, and remains functional underwater. The operation of overbraided magnetoresistive sensors underwater enable new application scenarios where sweat, rain, or totally submerged conditions can interfere with sensing of the device. Multiple of these overbraided magnetoresistive sensors were positioned within knitted channels of a textile sleeve to form a sensing surface, which demonstrated its capability as an interface for a virtual reality environment. Additionally, a overbraided magnetoresistive sensor was sewn onto a helmet strap to showcase its potential for safety applications.

## Results

The concept behind the touchless interactive magnetosensitive textile is illustrated in Fig. 1 (see also Supplementary Note 1). This textile was fabricated using overbraided magnetoresistive sensors which have the ability to bend, sheer and twist. These braided structures were created by incorporating flexible thin film magnetoresistive sensors[52,56–58] within its core. A close-up image of the sensing area of the magnetoresistive sensor is shown in Fig. 1a. The sensor layout features a meander structure with 20-µm-wide tracks, which increases the total resistivity, thereby reducing the electronic readout requirements. Thin film giant magnetoresistive (GMR) sensors were fabricated using fifty bilayers of 1 nm Co and 2.2 nm Cu deposited on a 50-µm-thick flexible polyimide foil (Supplementary Fig. S1). The fabricated thin-film GMR sensors exhibited excellent stability, with negligible change in performance after cyclic bending (Supplementary Fig. S2).

The GMR layer stack of alternating ferromagnetic and nonmagnetic metal thin films was coupled at the 2nd antiferromagnetic maximum and exhibits a large change in resistance when exposed to a small magnetic field[57]. This GMR layer stack also demonstrates consistent performance across different substrate materials, including Si wafers and various polymeric foils. Supplementary Fig. S3 compares the magnetoresistive performance of Co/Cu-based GMR multilayers deposited on different substrates. The impact of varying the number of Co and Cu bilayers on GMR sensor performance has been thoroughly investigated in a previous research[61]. For this study, we have used a GMR layer stack configuration that has been successfully employed in this prior investigation[61]. The homogeneity of the Co/Cu deposition was evaluated through transport measurements, indicating consistent GMR values across different radii of the substrates. This consistency suggests uniform deposition and sensor properties across the substrate area (Supplementary Fig. S14c). Additional confirmation of layer homogeneity was obtained through profilometer measurements using a Dektak device and this is presented in the Supplementary Fig. S4.

This structure was selected for its notorious magnetoresistive effect[57], primarily falling within a dynamic sensing range that stays below the 40 mT safety limit for magnetic fields, established by the World Health Organization (WHO)[62]. Additionally, we extend the technology also to anisotropic magnetoresistive (AMR) sensors based on 100-nm-thick $Fe_{19}Ni_{81}$ alloy that has increased sensitivity at smaller fields but possess overall smaller magnetoresistance (Supplementary Fig. S5).

There are three crucial steps (attached, encapsulation, and braided) during the fabrication process of overbraided magnetoresistive sensors (Fig. 1b and Supplementary Note 1). The overbraided sensors were initially fabricated by connecting individual flexible GMR sensors to litz wires using a conductive epoxy. Then, the flexible sensors are encapsulated using a UV curable polymer and positioned within a polyester textile braid. Polyester was chosen for the braid as it is the most used textile fibre globally[63,64]. The overbraided magnetoresistive sensor has an elliptical shape at the sensor location with the major axis having a length of $2.52 \pm 0.12$ mm and the minor axis having a length of $1.53 \pm 0.07$ mm. Figure 1c displays an image of the overbraided magnetoresistive sensor, revealing a section where the textile braid has been cut out to expose the embedded thin-film sensor. The performance of a GMR sensor when a magnetic field is applied perpendicular to the sensor element at each fabrication stage is presented in Fig. 1c. The sensor demonstrates a stable response when the magnetic field was varied at each fabrication stage (attached, encapsulation, and braided). The sensitivity changed only slightly during the overbraided magnetoresistive sensor manufacturing process. The individual resistance values of 15 overbraided magnetoresistive sensors were within the range of 100–1000 $\Omega$, when exposed to a magnetic field strength of 14 mT. The difference in resistance in between individual overbraided sensors is mostly due to manufacturing deviations and strain on the flexible sensors.

An important feature of wearable textiles is their drapabilty[65], which is the textile's ability to conform onto contours of a surface or body. However, the current drape testing methods and established standards, are not well-suited for evaluating the drape of E-textiles[66]. As a result, we opted for a bending rigidity test, which offered some insights into drape behaviour, even though it's not a direct measurement. To understand the impact of embedding sensors on the bending rigidity of the overbraided magnetoresistive sensor and textile, a Shirley stiffness test was conducted, and the results are displayed in Fig. 1e[67]. The data indicate that the bending rigidity of the textile braid increased by 23% after the integration of the magnetoresistive sensor. Nonetheless once the overbraided sensors were inserted into a fabric, the bending rigidity was measured at $(3155 \pm 437)$ mg cm along the vertical axis and $(1670 \pm 160)$ mg cm along the horizontal axis of the textile sleeve. This level of rigidity falls below that of certain woven structures crafted from heavy yarns[68]. While these magnetoresistive textiles may not be as flexible as traditional fabrics, they offer a reasonable level of conformability, ensuring that they don't compromise wearer comfort.

To showcase the capabilities of this technology, Fig. 1f illustrates an interactive system consisting of a textile glove embedded with a magnet and a knitted textile sleeve with four integrated overbraided magnetoresistive sensors. While overbraided magnetoresistive sensors can be directly woven into the sleeve, we opted for a knitted sleeve design with integrated channels. The overbraided sensors were positioned within these integrated channels. This design allowed for easier removal of the overbraided magnetoresistive sensors, facilitating separate recycling of the sensors and the fabric components. In the textile sleeve the sensing areas of the four overbraided magnetoresistive sensors were positioned $(30.1 \pm 1.7)$ mm (along the horizontal axis of the sleeve) and $(18.7 \pm 1.5)$ mm (along the horizontal axis of the sleeve) apart from each other. The glove contained a miniature magnet (3 mm diameter and 1 mm thickness) positioned on the index finger. Both the sleeve and the glove consist of fully wearable structures and were knitted seamlessly in a digital knitting machine. Detailed explanations of the structures can be found in Supplementary Notes 2, 3. Supplementary Fig. S11 displays images of the textile sleeve containing the overbraided magnetoresistive sensors conforming around a mannequin forearm. An experiment was conducted to evaluate the performance of the sleeve. Here the index finger was initially positioned away from the sleeve for 60 s. Then the finger was moved over sensor 1 for 60 s and then sensor 2 and so forth. Lastly, the finger was taken away from the sleeve for the final 60 s. It can be observed in Fig. 1g that the overbraided magnetoresistive sensors are able to

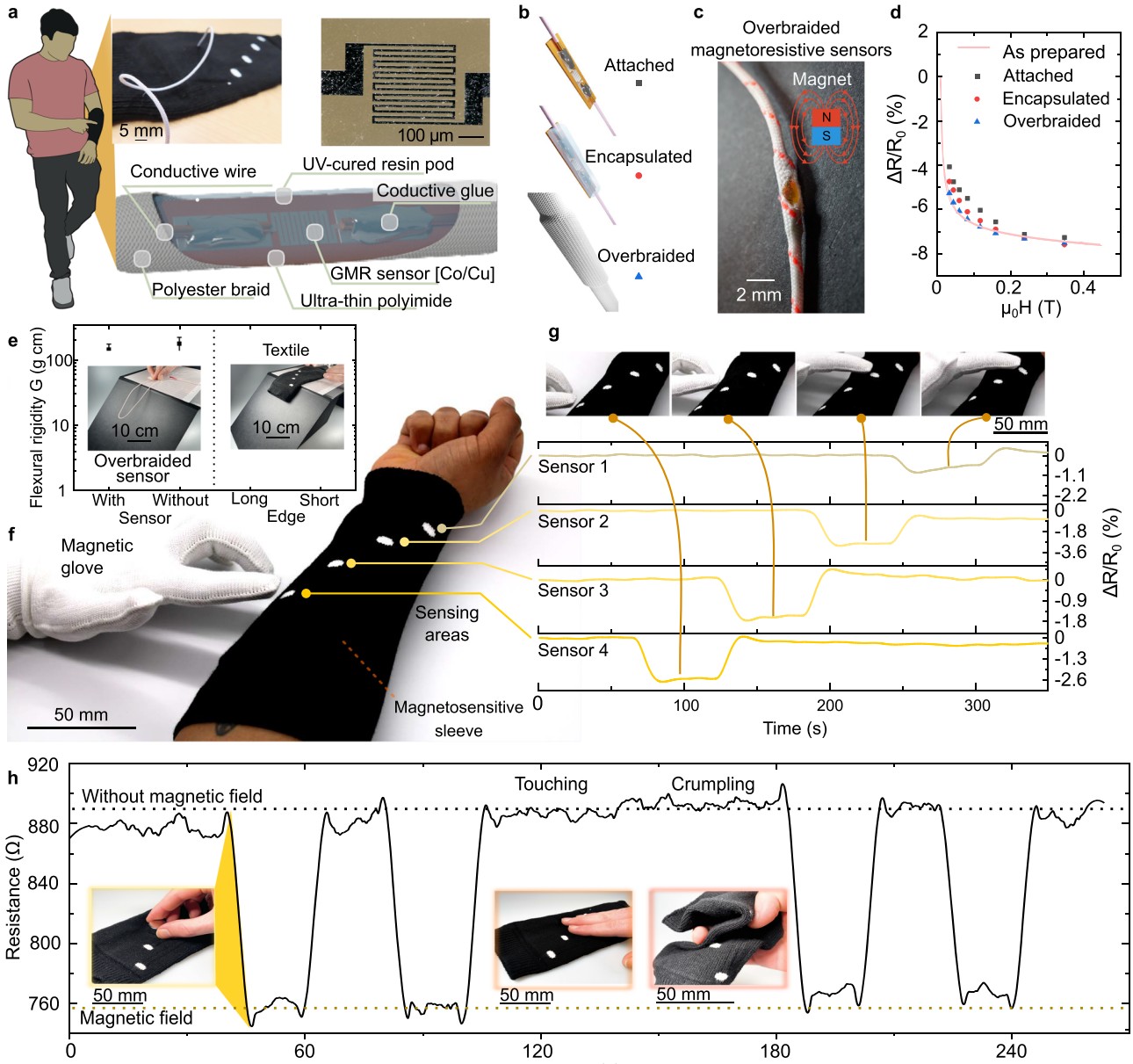

**Fig. 1 | Concept of a magnetoresistive textile. a** The concept of highly flexible overbraided magnetoresistive sensor involves embedding a miniature flexible magnetoresistive sensor within the core of a textile braid. **b** The fabrication stages of the overbraided magnetoresistive sensor. First, the sensor is attached to litz wires, followed by encapsulation, and integration within a textile braid. **c** Photograph of a overbraided magnetoresistive sensor with an opening to reveal the embedded sensor within. **d** Performance of a representative GMR sensor at various fabrications stages when subjected to varying magnetic fields perpendicular to the sensor element. Data represents eight discrete measurement steps. **e** Measurements from the Shirley stiffness test, which indicate the bending rigidity of both the overbraided magnetoresistive sensor and a textile fabricated using them. **f** An interactive textile system created using a knitted sleeve containing four overbraided magnetoresistive sensors and a glove embedded with a miniature magnet. **g** Results from an experiment where the finger of the glove containing the magnet is positioned initially away from the sleeve thereafter above each of the four sensing areas. **h** Measurements from a overbraided magnetoresistive sensor obtained before, during and after a mechanical manipulation (crumpling and touching) with the sleeve.

accurately distinguish when the finger within the glove approaches each of the sensors. The sensors are activated without any relevant interference or cross talk between them. The suitability of the sensor to operate at frequencies up to 10 kHz is demonstrated in the sensor's frequency response given in Supplementary Fig. S13. This demonstrates its suitability for real-time interactive applications. The knitted sleeve demonstrates key textile properties such as bending, and sheering. Figure 1h demonstrates the performance of overbraided magnetoresistive sensors while they were being touched and crumpled. Supplementary Video 1 demonstrates the ability of our smart textiles to avoid accidental touch activations and deformations.

Scaling up the production of several metres of this braid (Fig. 2a) can be achieved by fabricating magnetic field sensors on large-area flexible substrates, up to 300 mm in diameter (Supplementary Fig. S14). With a sensor footprint of 1 cm², a 300 mm wafer, which has an area of about 70,650 mm², can accommodate roughly 7000 sensors. Assuming one sensor is placed per centimetre of braid, this allows for the production of approximately 70 m of braid from a single substrate wafer. Their stable magnetoresistance output upon deformation (Supplementary Fig. S15) enables the reliable sensing during mechanical deformations of smart textiles. An analysis of the overbraided magnetoresistive sensor's performance was conducted for

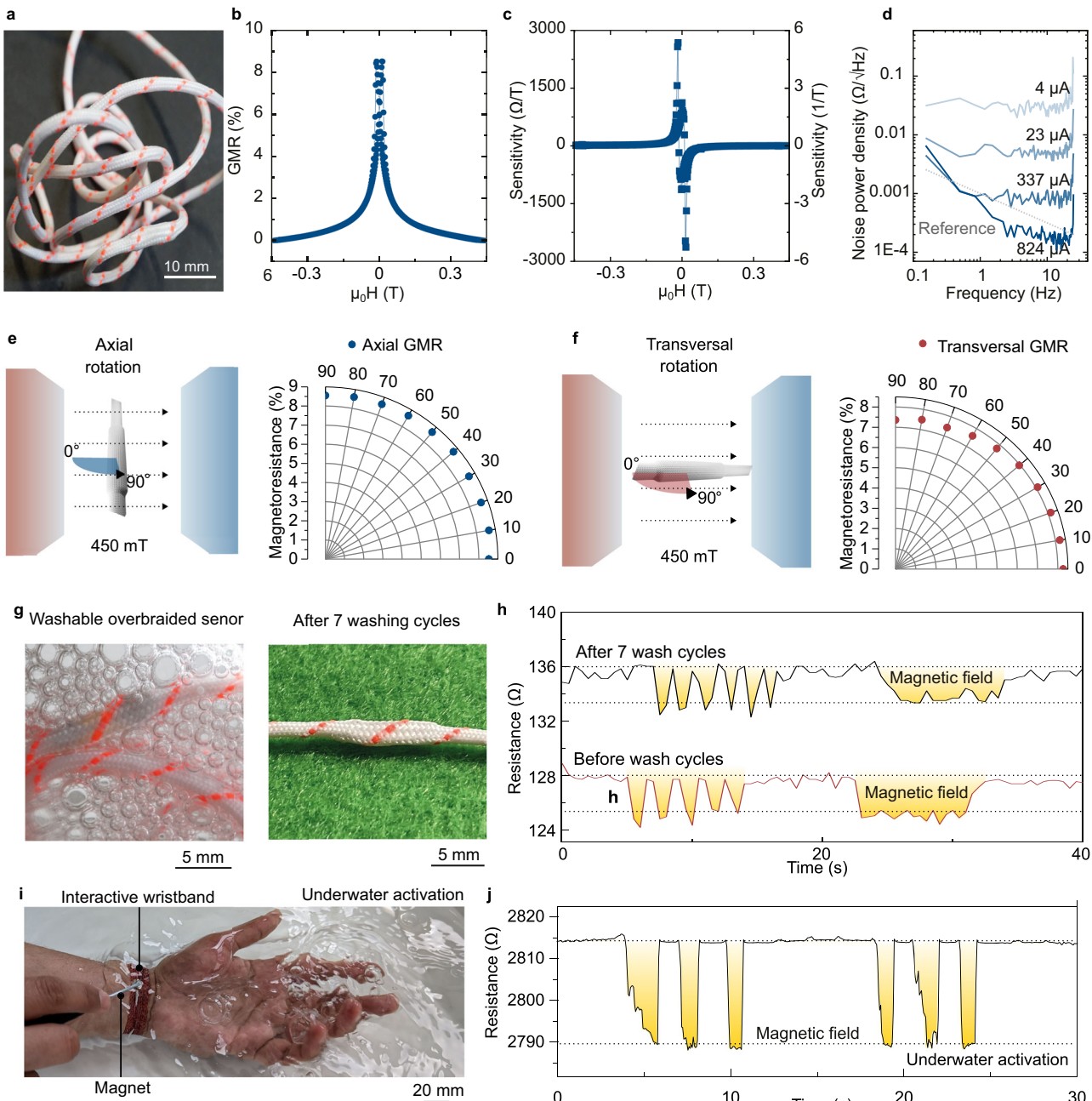

**Fig. 2 | Performance of overbraided magnetoresistive sensors. a** Photograph of a long overbraided magnetoresistive sensor that is highly flexible (bendable and twistable). **b** GMR response obtained from a overbraided magnetoresistive sensor exposed to an external magnetic field. **c** The sensitivity of a representative over-braided magnetoresistive sensor and (**d**) the noise power density of the signal from the overbraided sensor after applying different current levels. The full GMR response up to saturation at about 450 mT of a overbraided magnetoresistive sensor showed isotropic response amplitude upon rotations of the sensor (**e**) along and (**f**) transverse to the main symmetry axis. **g** Photograph of a overbraided magnetor-esistive sensor during washing and after seven wash cycles in a washing machine. **h** Response of overbraided magnetoresistive sensor to an approaching permanent magnet before and after seven machine wash cycles. **i** Activations of a smart wristband in underwater conditions. **j** The overbraided magnetoresistive sensor is activated reliably in underwater conditions when a permanent magnet is approached.

different magnetic field strengths. Here an external magnetic field was applied along the main axis of the braid and the response of the overbraided magnetoresistive sensor was recorded (Fig. 2b). Being consistent with the behaviour of the GMR multilayer systems, the resistance decreased with an increasing magnetic field strength and tappered off for magnetic field strengths of >100 mT. The overbraided magnetoresistive sensor had a maximum sensitivity at 17 mT (Fig. 2c) and exhibited noise levels down to 753 μΩ Hz$^{-½}$ (Fig. 2d). Based on these sensor parameters, a sensor limit of detection of 380 nT was achieved for overbraided magnetoresistive sensors

based on GMR effect. Since there is limited information of other textile-integrated magnetic field sensors, we compared our overbraided magne-toresistive sensors performance with other flexible magnetic field sensors used for interactivity. This comparison is provided in Supplementary Table S2. The analysis of the AMR sensors can be found in Supplementary Fig. S16. Relevant for proximity measurement applications in smart textiles, the change of magnetoresistance was close to isotropic for magnetic fields applied from different angles along (Fig. 2e) and transversal (Fig. 2f) to the braid axis. This outcome represents an appealing characteristic achieved

following the incorporation of the sensors within the overbraid, since the GMR effect in thin film is known to be barely sensitive to the out-of-plane magnetic fields. When applied practically, the employed permanent magnet can be brought near the overbraided magnetoresistive sensor from any orientation, and the sensor will show the same total amplitude change regardless of the direction from which it is approached. Still, when the full curves are analysed, they show a slight degradation of sensitivity upon transversal rotations (Supplementary Fig. S17). On the other hand, overbraided magnetoresistive sensors based on the AMR effect show strong anisotropy with respect to the direction of the applied field making them good candidates to be applied for directional sensing (Supplementary Fig. S18).

Next, the impact of washing on the overbraided magnetoresistive sensor's performance was studied, based on the method proposed by Rotzler et al.[69]. This method exerts a high amount of stress on the textile. Overall, seven wash cycles were performed. After each cycle, the measurement procedure mentioned in the methods section was carried out. There was no visible damage to the overbraided sensor during and after the washing procedure (Fig. 2g). The results from before and after the wash tests are given in Fig. 2h. The results indicate that the overbraided magnetoresistive sensors response to an approaching magnet is not altered by the wash test at all. The sensor before and after wash testing was able to measure when a magnetic field was directed towards it for 1 s and then withdrawn for 1 s in a repetitive manner. In addition, the sensor was able to detect when it was exposed to a magnetic field for 10 s. After the washing process, there is a slight 6.4% rise of the absolute sensor resistance when no magnet is in its proximity. This change may be due to changes in resistance of the litz wire or the contacts during the washing process.

Wearable textiles must function in everyday conditions where humidity is a factor. They could be even submerged in underwater environments like when the user exercises or during a rainy day. Avoiding losing functionality in these common activities is crucial to provide reliable interfacing. Magnetic fields are pervasive in humid environments and virtually immune to the change of medium from ambient air to submerged in a water bath (Fig. 2i). We embedded a overbraided magnetoresistive sensor into a smart wristband to test their performance in underwater conditions (Supplementary Video 2), where other non-contact capacitive interfaces fail. The resistance changes of the overbraided magnetoresistive sensor when a permanent magnet is approaching the smart wristband submerged in tap water is significantly higher than the signal changes caused by unwanted taps and the normal motion of the sensing element during wear (Fig. 2h).

The potential of these overbraided magnetoresistive sensors to add a new dimension to everyday clothing is demonstrated using two applications. Figure 3a showcases a knitted sleeve used to navigate through a virtual reality environment. Supplementary Video 3 shows the sleeves functionality in real time, while the graph in Fig. 3b illustrates the resistance changes of two of the overbraided magnetoresistive sensors (utilized for turning left and moving forward illustrated in Fig. 3c) when approached by a magnetic ring. Figure 3d showcases another use case scenario. Here the overbraided magnetoresistive sensor is used in a strap of a magnetic buckle on a helmet (Fig. 3e) and can detect whether the helmet is secured on. Supplementary Video 4 demonstrates the functionality of the helmet. The change of the sensor resistance of the overbraided magnetoresistive sensor on the helmet strap is displayed in Fig. 3f. As shown in Fig. 3g, the screen attached to the helmet serves as a safety indicator by determining whether the helmet is securely fastened or not based on the input of the safety strap.

## Discussion

This paper introduced a new technique of incorporating flexible magnetoresistive sensors within the core of a textile braid. It is the first time where magnetic sensitivity has been successfully incorporated into smart textiles in a mechanically resilient and conformal way, making them suitable for everyday use. The 1D overbraided magnetoresistive sensors and textiles produced from these overbraided sensors exhibit the ability to bend,

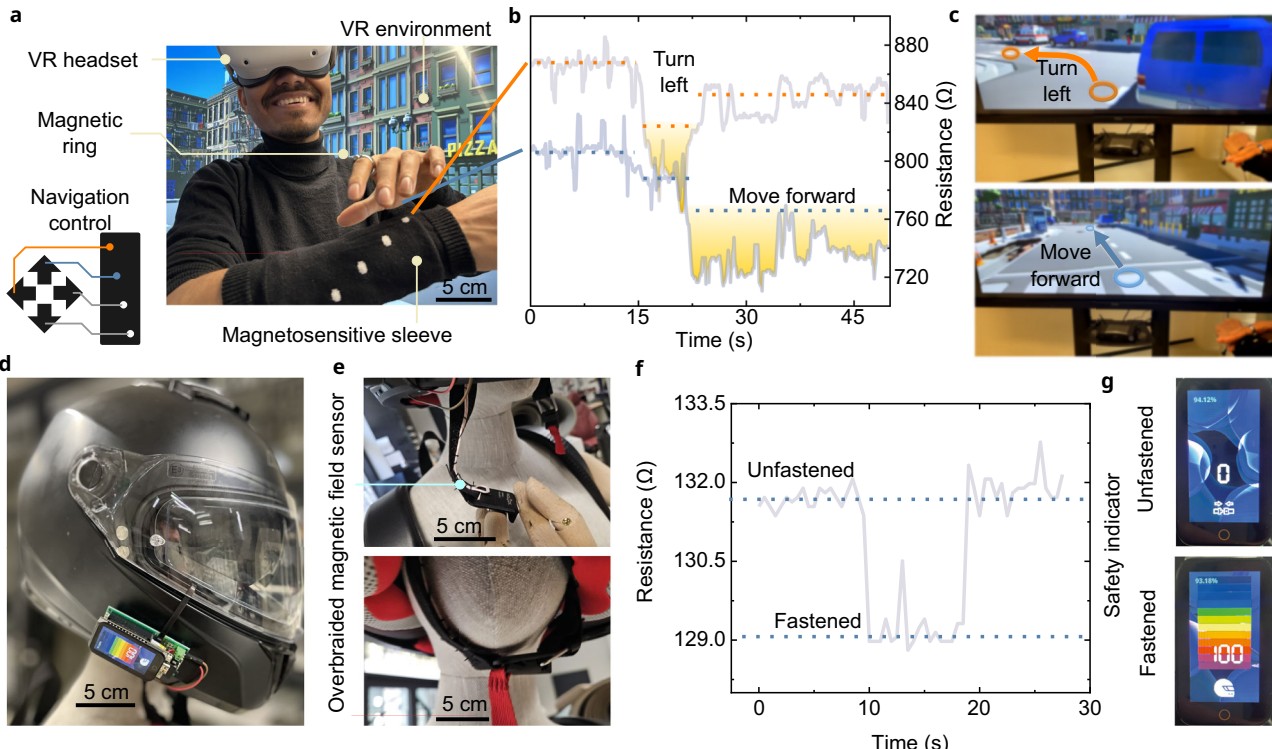

**Fig. 3 | Use case scenarios for overbraided magnetoresistive sensor embedded textiles. a** A virtual reality environment navigable using the knitted sleeve and a magnetic ring. **b** The measurements from two overbraided magnetoresistive sensors when they are used to navigate the VR environment. **c** The two overbraided magnetoresistive sensors in the sleeve were utilised to move forward and turn left. **d** A helmet equipped with a magnetic buckle and a overbraided magnetoresistive sensor stitched onto its strap as shown in a close-up in (**e**). **f** Measurements obtained from the overbraided magnetoresistive sensor stitched on the helmet strapped on or off. **g** The screen indicates whether the helmet is strapped on or off.

undergo sheer and to be washed in a conventional washing machine without deterioration in their sensing performance. The overbraided magnetoresistive sensors also remained functional underwater. Interactive textiles created using magnetic field sensors demonstrate enhanced selectivity and would eliminate false activations in comparison to previously reported touchless interactive textiles. Moreover, the overbraided magnetoresistive sensor is isotropically activated with magnetic fields from different orientations which is an attractive trait, since the permanent magnet can be brought near the overbraided magnetoresistive sensor from any orientation. This will be the case when these overbraided sensors are incorporated into textile and used in everyday scenarios. Two demonstrators were developed utilizing these overbraided magnetoresistive sensors: A knitted sleeve capable of serving as a VR environment interface, and a helmet strap designed to detect if it is securely fastened.

The implementation of touchless magnetic interactivity into wearable and interactive applications extends beyond entertainment, establishing its relevance in safety-critical contexts. This includes protective suits, underwater garments, and situations where gloves are in use such as surgery rooms and industrial settings. The power requirements of these sensors are compatible with energy harvesters developed for wearable textiles[29,30], making them fully operational as textile magnetic interfaces. This integrations strategy can be extended to other functional magnetosensitive elements like AMR sensors, where their high sensitivity to small fields and anisotropic behaviour can be potentially used to detect motion activity, biological signals and orientation.

## Methods
### Fabrication of the flexible GMR sensors
Flexible magnetoresistive sensors were fabricated on polyimide foils (HD Microsystems, USA) with a thickness of 50 μm. The foils were temporarily fixed to a PDMS (polydimethylsiloxane, Sylgard 184, Dow Corning) coated $5 \times 5$ cm$^2$ glass slide. The patterning of meander structures was done via direct laser writing (DWL 66 Laser Writer, Heidelberg Instruments, Germany) using AZ5214e (MicroChemicals GmbH, Germany) resist. The photoresist was spin-coated at 4000 rpm, and soft baked at 110 °C for 60 s. After exposure, the sample was post-baked at 120 °C for 120 s, flood exposed with a UV lamp for 30 s (proMa 140 017, Germany) and developed using a 1:4 solution of AZ351B (MicroChemicals GmbH, Germany) and deionized water. Ar sputtering deposition was employed to fabricate multilayers of 50 bilayers of [Co (1 nm)/Cu (2.2 nm)], coupled at the 2$^{nd}$ antiferromagnetic maximum (Ar pressure: $10^{-3}$ mbar, base pressure: $10^{-7}$ mbar, deposition rate: 2 Å s$^{-1}$). The liftoff of the sensors was carried out in an acetone bath removing the remaining photoresist. We also prepared AMR sensors (Supplementary Figs. S5, S16, S18) using identical photolithohraohy process and deposition via magnetron sputtering of 100 nm of Fe$_{19}$Ni$_{81}$ (Ar pressure: $8 \times 10^{-4}$ mbar, base pressure $10^{-7}$ mbar, deposition rate 3 Å s$^{-1}$). Tests for large area fabrication of these sensors (Supplementary Fig. S14) where done on 100 mm diameter flexible substrates (125-μm-thick PET, Melinex ST 504, Dupont Teijin Films, USA), and polyimide foil (50-μm-thick, HD Microsystems, USA).

### Fabrication of overbraided magnetoresistive sensors
Overbraided magnetoresistive sensors were fabricated by attaching the flexible sensors onto litz wires (outer diameter: 254 μm[70], resistivity: $1.329 \pm 0.002$ Ω/m, BXL2001, OSCO Ltd., Milton Keynes, UK). The ends of litz wires had their enamel and fibre coating removed using a contact soldering iron (Antex XS25; Antex Electronics Limited, Plymouth, UK), and a small amount of solder was applied (SA305 solder wire; RS Components Ltd., Corby, UK). The wires were adhered to the magnetoresistive sensors using CW2400 conductive epoxy (Chemtronics, Kennesaw, Georgia, USA) as illustrated in Supplementary Fig. S7. Once attached, the sensors and interconnects were encapsulated using a UV curable encapsulant (Dymax 9001-E-V3.5; Dymax Corporation, Torrington, CT, USA) as displayed in Supplementary Fig. S8. These devices were then positioned within a braided tube with 4 polyester yarns (48 filaments/150 denier). The covering braided

tube consisted of 24 carriers with polyester yarns ($2 \times 36$ filaments/167 dtex; Ashworth and Sons, Cheshire, UK) and was produced using a suture braiding machine (RU1/24-80, Herzog GmbH, Oldenburg, Germany) as shown in Supplementary Fig. S9.

### Magnetoresistance characterization
For the experiments shown in Fig. 2b–d, overbraided magnetoresistive sensors were placed in the magnetic field of an electromagnet to register their resistance changes. The electromagnet created a homogeneous field up to 450 mT. A Tensormeter measurement unit (HZDR Innovation GmbH, Germany) was applied to measure the longitudinal resistance of the overbraided sensor using a 2-point configuration. The employed current and frequency for characterization was between 1 μA and 1 mA and 1775 Hz, respectively. The magnetoresistance (MR) was calculated as MR = 100% $(R - R_{sat})/R_{sat}$ and was measured with the overbraided sensors oriented along the braid main axis (axial direction), as well perpendicular to it (transversal direction). A rotating holder allowed to measure the MR response at different angles with respect to these rotation axes (Supplementary Figs. S17, S18). A 3D printed bending holder with clamps with tunable distance was used to measure the magnetoresistance response upon different bending radius (Supplementary Fig. S15). The bending performance measurements were carried along the current directions of the sensors.

For the experiments shown in Fig. 1d a bespoke Perspex rig was designed. The design was made such that a N42 permanent magnet (16.5 kg pull; 20 mm thickness, Magnet Expert LTD, Newark, UK) was placed at the bottom of the rig in an enclosed area. The height (distance) between the samples and the magnet was increased by adding Perspex plates in between them (the plates included a whole larger than the sensor to ensure that the rig did not interfere with readings). Measurements were taken at different heights. At each height the samples were left for 15 s to reach a steady state prior to taking the measurements. A DAQ6510 data acquisition/multimeter system (Keithley, Cleveland, Ohio, USA) was used to measure the resistance of the samples. The magnetic field strength at each height from the magnet was measured using a gauss metre GM07 (Hirst Magnetic Instruments Ltd, Falmouth, UK).

### Fabrication of the prototype textile garments
The seamless jacquard sleeve was knitted on a Stoll CMS ADF 32 W E7.2 (Reutlingen, Germany) digital knitting machine with a rib structure on both ends to ensure a snug arm fit. The 3D structure of the sleeve was knitted in one seamless process, and it is explained in Supplementary Note 2. As shown in Supplementary Fig. S10, four channels were created on the knitted sleeve to incorporate the four overbraided magnetoresistive sensors. A fluid yarn (1PLY 2% Elastane, 7% Nylon, 91% Viscose, Yeoman Yarns, Leicester, UK) was used to knit the sleeve. The knitted seamless glove that contained the magnet (N52 permanent magnet, diameter 3 mm and thickness 1 mm, surface magnetic field: 4105 G, pull force: 0.14 kg, Magnet Expert LTD, Newark, UK) was also created on the Stoll CMS ADF 32 W E7.2 machine. Similar to the sleeve, the glove was knitted in one seamless process as further detailed in Supplementary Note 3. Three different types of yarn were used for the glove: fluid yarn (1PLY 2% Elastane, 7% Nylon, 91% Viscose, Yeoman Yarns), cotton yarn (1/24 combed cotton 1 ply, Yeoman Yarns) and Nylon 2/78/68 (CONTIFIBRE S.P.A., Tobagi, Italy). The glove was knitted with a channel for a miniature N52 permanent magnet as illustrated in Supplementary Fig. S12.

### Wash test procedure
The wash test was conducted using a Bosch TitanEdition Series 6 (Gerlingen, Germany) washing machine. A neutral soap and 1 kg of support fabric was used in every wash cycle. The washing machine was set to 800 rpm, with a water volume of 12 L, and temperature of 30 °C, using 1% by weight of soup, and washed for 35 min, followed by a spin cycle (2 min at 500 rpm and 5 min at 800 rpm), and rinsing (3 min, 20 °C, 40% on-time, 12 L of water – 3 min, 20 °C, 40% on-time, 12 L of water). All tests were performed at ambient conditions (ambient relative humidity). After each washing cycle,

**Article**

the sample was air dried at room temperature for 30 min. To test the sample, the following measurement procedure was performed following each wash cycle: Initially for 5 s, there was no magnet near the overbraided magnetoresistive sensor. Subsequently, within the next 10 s, the magnet with a surface flux density of 0.414 T was brought towards the overbraided sensor for 1 s, then removed for 1 s, and this process was repeated for 10 s. Afterward, the magnet was moved away for 10 s. Following this, the magnet was brought close to the overbraided sensor and held in place for 10 s, followed by a withdrawal for an additional 10 s.

## Shirley stiffness test procedure

The bending rigidity, $G = M\,L^3$, was determined by measuring the textile's specific weight per unit area, $M$, and the bending length $L$, as defined by the Shirley stiffness test[71]. This is an established method to quantify the stiffness of smart textiles. Here, the overbraided magnetoresistive sensors and the fabric were manually moved over a horizontal plane towards an edge with a declining ramp at the angle of 41.5°. As the samples' ends reached the horizontal plane's edge, they begin to bend downward due to gravity. The moment the textile tip touches the angled plane, movement is halted, and the length of the bent textile, $L$, is measured. All measurements were repeated 5 times and the average was obtained.

## Underwater demonstrator

For the underwater demonstrator (Supplementary Video 2), a Tensormeter measurement unit (HZDR Innovation GmbH, Germany) was used to track the resistance changes of a overbraided magnetoresistive sensor. The sensor biased with 2 V and 1 mA was measured at a rate of 200 ms. The overbraided magnetoresistive sensor was integrated to a wristband. The interconnections from litz wires to BNC connectors of the Tensormeter measurement unit were done in air. A plastic container was filled with tap water to test the touchless activation when the wristband was submerged and approached by a permanent magnet (300 mT at the surface). The data was acquired and indicated through a visual interface developed in LABView software (National Instruments, USA).

## Virtual reality (VR) simulator

For the VR demonstrator (Supplementary Video 3), an Adafruit HUZZAH ESP32 board (New York, USA), with a 12 bit ADC for sampling the voltage dividers was used to interface with the sensors. A reference resistor with a value of $R_{ref} = 350\,\Omega$ was used and connected to a 3.3 V source through the four GPIO pins on the ESP32 board. To reduce measurement noise, we used the Exponentially Weighted Moving Average filter with:

$$S_i = \begin{cases} X_i & if\ i = 1 \\ \alpha X_i + (1-\alpha)S_{i-1} & if\ i > 1 \end{cases}$$

Readings were gathered within 100 ms. This was then sent over the OSC protocol to a Unity 3D application controlling the locomotion.

## Fabrication of the prototype helmet

For the helmet strap demo (Supplementary Video 4), we used a Fidlock magnet snap (Hannover, Germany) for closing the helmet combined with the overbraided magnetoresistive sensor. For sensing the signal, a LilyGO ESP32-S3 board (Shenzhen Xinyuan Electronic Technology Co., Shenzhen, China) was chosen, using the on-board 12 bit ADC for sampling. For the voltage divider a reference resistor with a value of $R_{ref} = 500\,\Omega$ was used and connected to a 3.3 V source through a separate GPIO pin on the ESP32, with their configurations defined in the firmware to regulate the flow of current. The signal was filtered using the Exponentially Weighted Moving Average filter.

## Ethics

We applied a permanent magnet to a finger of a user in several studies reported in this manuscript. This magnet is not an electronic component. Furthermore, in several studies, a sensor is applied on skin of a user (two

volunteers; male). These studies are done accordingly to the ethic approval #SR-EK-459122024 from the ethics committee at the Technical University of Dresden. We have written consents from volunteers to participate in this study including the publication of identifiable images of the research participant. The current lines are always isolated and are not in touch with the skin. The sensor is not worn on skin for any extended duration.

## Data availability

The data supporting the findings of this study are available within the main text of this article and its Supplementary Information. Further information on this study is available from the corresponding authors upon reasonable request.

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

## Acknowledgements

We thank Conrad Schubert and Dr. Gilbert Santiago Cañón Bermúdez (both HZDR) for their support with the fabrication of flexible sensors. We thank Arash Shahidi for his assistance with fabricating overbraided magnetoresistive sensors. We acknowledge Lisa Mussner for support in modelling of the scene for VR demonstration. We thank Dr. Mykola Vinnichenko (Fraunhofer IKTS Dresden) for providing polymeric substrates. This research was funded by the Engineering and Physical Sciences Research Council (EPSRC) grant EP/T001313/1 Production Engineering Research for the Manufacture of Novel Electronically Functional Yarns for Multifunctional Smart Textiles. The core technique of manufacturing the overbraided sensors are covered by a portfolio of patents including US20190003084A1, US10,301,751B2, GB 2529900B, EP3191632B1, and CN106715769B. This work is financially supported in part by the German Research Foundation (DFG) grants MA 5144/28−1, European Commission HORIZON RIA (project REGO; ID: 101070066) and ERC grant 3DmultiFerro (Project number: 101141331).

## Author contributions

P.L. and E.S.O.-M. contributed equally to this work. P.L. and N.M. developed the concept and initiated the collaboration. E.S.O.-M., M.H., and D.M. designed, fabricated and characterized flexible magnetic field sensors. C.O., K.M., P.L., N.M. M.H., T.D., and T.H.-R. provided the expertise and design of braiding, textile knitting fabrication and characterization. M.H., R.B., N.P., P.L., E.S.O.-M. and D.M. worked on the implementation of the demonstrators. P.L., E.S.O.-M., K.M., D.M. prepared figures and wrote the manuscript, and T.H.-R., M.H., N.M. and D.M. edited and revised the manuscript. T.H.-R., N.M., and D.M. supervised the project. All of the authors discussed the results and commented on the paper.

## Funding

## Competing interests

The authors declare no competing interests.
