## [Peer Review file · Communications Engineering]

Submersible touchless interactivity in conformable textiles enabled by highly selective overbraided magnetoresistive sensors

Corresponding Author: Dr Denys Makarov

Version 0:

Reviewer comments:

Reviewer #1

(Remarks to the Author)

This work prepares a magnetically sensitive yarn for non-contact interactive textiles. It is a good job, especially the application presentation part. Here are a few suggestions to consider before publishing.

1. There are many flexible magnetic-based smart wearable textiles that “bridges the integration gap between on-skin and rigid magnetic interfaces” so what is the key significance of this work?
2. These smart yarns exhibit a detectivity down to 380 nT. So is there more literature that needs to be compared to prove the high sensitivity of this yarn?
3. Is it necessary to do more gradient experiments to verify the optimal formulation?
4. The content of Table 1 is suggested to include performance comparisons.

Reviewer #2

(Remarks to the Author)

In this article, the authors presented 1D magnetosensitive yarns that exhibits good detectability and nearly isotropic magnetoresistance amplitude response, possesses good stability against water washing and stability against mechanical signal interference. The magnetosensitive yarn can be used underwater, has a high potential for application in the field of touchless interactive interaction. The paper is well written for easy understanding, and also provided satisfactory technological details related to the design of the GMR sensor. I would recommend the publication of this paper after the follow concerns being addressed.

1. In the process of GMR sensor preparation, whether the number of layers of deposited layers of Co and Cu deposited on the polyimide film will affect the performance of the sensor.
2. Could the deposition homogeneity of the Co and Cu be validated by cross-sectional SEM characterization?
3. The authors measured the change during bending of GMR sensors to a radius of 4 mm. Considering that the sensors might experience bending repeatedly, the sensing stability was suggested to characterize under cyclic bending events to verify the device durability?
4. The response time of the GMR sensor should be tested.
5. The GMR sensor shows excellent performance even underwater, a practical application scenario was suggested to demonstrated to show the advantage over other sensors.
6. In the Introduction part, the author summarized the yarn/textile based electronics for comparison, some highly relevant references were suggested to be added for a more comprehensive background (Advanced Materials Technologies 5.1 (2020): 1900781; Advanced Functional Materials 33.44 (2023): 2303881; ACS nano 8.5 (2014): 4571-4579; Advanced materials, 2015, 27(45): 7365-7371).
7. The first paragraph of the Discussion part should be included into the Results part?
8. The number of the Supplementary figures seems out of order, please confirm it.

Reviewer #3

(Remarks to the Author)

Please see attached

Version 1:

Reviewer comments:

Reviewer #1

(Remarks to the Author)

Reviewer #2

(Remarks to the Author)

The authors have well addressed the questions, I recommend the publication as it is.

Reviewer #3

(Remarks to the Author)

Please see attached.

Response letter

We thank the reviewers for their insightful comments, which have helped to improve the quality of the manuscript as well as allowed us to clarify several aspects of this work. We have revised the manuscript in accordance with the comments provided and have highlighted the changes. Furthermore, we have included the changes as extracts under each point along with reasoning or further explanation as needed, below.

For clarity, the reviewer's comments have been left as black text, our responses have been written in blue text, and any information added to the manuscript has been highlighted as red text. The additional elements have been highlighted in the manuscript as well to make it easier to see where the additions have been included.

Reviewer #1:

This work prepares a magnetically sensitive yarn for non-contact interactive textiles. It is a good job, especially the application presentation part. Here are a few suggestions to consider before publishing.

1. There are many flexible magnetic-based smart wearable textiles that “bridges the integration gap between on-skin and rigid magnetic interfaces” so what is the key significance of this work?

To demonstrate the novelty of our work as the first to build and utilize washable magnetically sensitive yarns, we have assembled a comparison table (Table S1, Supplementary Information) that showcases existing magnetically sensing elements integrated into textiles. This table highlights the lack of comparable research using yarns as the core building block for interactive textiles.

We have added the following text to the manuscript:

1/ Introduction

Supplementary Table S1 presents a comparison of our magnetic sensing technology against other available magnetic sensing technologies in textiles.

2/ We have added the following table to the supplementary information

Ref	Sensor Type	Integration Method	Sensitivity/LOD	Sensor Form Factor
[1]	Magnetic Eddy Current Induction	Coils sewed into a shirt	Not specified	Wire (coil)
[2]	Hall Effect	Integrated with conductive woven fabric	Not specified	Rigid (ESP32 module)
[3]	Dual-mode: Magneto-straining	Coating on spandex substrate	0.7 kPa ⁻¹ (pressure), 13.76 mm ⁻¹ (proximity)	2D coating Film
[4]	Dual-mode: Strain & Magnetic Field	Dropping-drying process	44% resistance change at 8% strain, 1.5% relative resistance increment at 44° bending under 400 mT	1D Coaxial structure
This work	Giant Magnetoresistive (GMR)	Embedded within braided textile yarns	380 nT, > 7% relative resistance increment at 450 mT	1D Yarn (braided)

Table S1: Comparison of magnetic sensing technologies for electronic textiles, highlighting the first successful implementation of sensing at the yarn level, enabling conformability and performance in knitted fabrics.

2. These smart yarns exhibit a detectivity down to 380 nT. So is there more literature that needs to be compared to prove the high sensitivity of this yarn?

Since there is limited information of other textile-integrated magnetic field sensors, we compared our magnetoresistive electronic yarns with other flexible magnetic field sensors used for interactivity. To address this, we have added a new table (Table S2, Supplementary Information) summarizing the sensitivities and limits of detection of various flexible magnetic field sensors, including both AMR and GMR technologies. This table demonstrates the competitive performance of our yarns compared to existing flexible sensor technologies used in interactive applications.

We have added the following text to the manuscript:

1/ Results

Since there is limited information of other textile-integrated magnetic field sensors, we compared our magnetoresistive electronic yarns performance with other flexible magnetic field sensors used for interactivity. This comparison is provided in Supplementary Table S2.

2/ We have added the following table to the supplementary information

Ref	Sensor Type	Sensor Form Factor	Integration Method	Sensitivity/LOD
[13]	Magnetoelectric (ME)	Flexible Laminate	Bonding of Metglas foils to PZT thick film on mica substrate	200 nT at low frequencies, 200 pT at resonance
[14]	Hall Effect	Flexible (Laser-Scribed Graphene)	Laser scribing on polyimide substrate	1.12 V/AT, 0.446 mT/Hz resolution
[15]	Anisotropic Magnetoresistive (AMR)	3D Micro-Origami Cubic Architecture	Self-assembly of micro-origami cubes on a wafer-scale, integrated with a-IGZO TFTs	$(0.068 \pm 0.022) \text{ T}^{-1}$ for three orthogonal axes
[16]	Anisotropic Magnetoresistive (AMR)	Printed, Flexible	Printed onto flexible substrates	35.7 T^{-1} at 86nT, 36 nT resolution
[17]	Anisotropic Magnetoresistive (AMR)	Flexible	Deposited on ultrathin Kapton substrate	0.25 Oe^{-1}
[18]	Giant Magnetoresistive (GMR)	Printed, Flexible	Printed on ultrathin polymer foils	3 T^{-1} at 0.88 mT
This work	Giant Magnetoresistive (GMR)	1D Yarn (braided)	Embedded within braided textile yarns	380 nT resolution, >7% relative resistance increment at 450 mT

Table S2: Comparison of flexible magnetic sensor technologies for interactive applications. While previous works have focused on thin-film or printed sensor formats, this work introduces a novel approach utilizing a 1D yarn form factor for GMR sensors. This strategy results in highly sensitive and comfortable interactive textiles, enabling new possibilities for seamless integration of magnetic sensing into fabrics.

3. Is it necessary to do more gradient experiments to verify the optimal formulation?

We performed a series of deposition of nominally same layer stack on different polymeric substrates and determined that the GMR performance of ~7% is very robust when the deposition is done on a broad range of polymer materials.

We have added this explanation to the manuscript:

1/ Results

This GMR layer stack also demonstrates consistent performance across different substrate materials, including Si wafers and various polymeric foils. Supplementary Figure S3 compares the magnetoresistive performance of Co/Cu-based GMR multilayers deposited on different substrates.

2/ We have added the following figure to the supplementary information

Supplementary Figure S3. Comparison of the magnetoresistive performance of Co/Cu-based GMR multilayers deposited on different flexible substrates.

4. The content of Table 1 is suggested to include performance comparisons.

We have addressed this comment by adding new tables reported as Supplementary Tables S1 and S2. These tables compare our work with existing technologies.

Reviewer #2:

In this article, the authors presented 1D magnetosensitive yarns that exhibits good detectability and nearly isotropic magnetoresistance amplitude response, possesses good stability against water washing and stability against mechanical signal interference. The magnetosensitive yarn can be used underwater, has a high potential for application in the field of touchless interactive interaction. The paper is well written for easy understanding, and also provided satisfactory technological details related to the design of the GMR sensor. I would recommend the publication of this paper after the follow concerns being addressed.

1. In the process of GMR sensor preparation, whether the number of layers of deposited layers of Co and Cu deposited on the polyimide film will affect the performance of the sensor.

The number of Co/Cu bilayers does influence the GMR sensor's performance. The relationship between the number of bilayers and GMR sensor performance has been extensively studied in the literature and we used a stack from a previous study (DOI-10.1002/adma.202005521). Increasing the number of bilayers generally enhances the GMR effect until saturation but this can also increase the fabrication time.

We have added this text to the manuscript:

1/ Results

The impact of varying the number of Co and Cu bilayers on GMR sensor performance has been thoroughly investigated in a previous work.^[61] For this study, we have used a GMR layer stack configuration that has been successfully employed in this prior investigation. ^[61]

2. Could the deposition homogeneity of the Co and Cu be validated by cross-sectional SEM characterization?

Following the remark of the reviewer, we performed measurement of the homogeneity of the deposited layer thickness using a Dektak profilometer (please see the figure below). This is a standard and robust method, which we use to optimise thicknesses of our magnetoresistive sensors. Furthermore, the already reported magnetotransport measurements (Supplementary Figure S14) clearly reveal consistent GMR values across different radii on the substrates. This consistency suggests uniform deposition and sensor properties across the substrate area.

The following text has been added to the manuscript.

The homogeneity of the Co/Cu deposition was evaluated through transport measurements, indicating consistent GMR values across different radii of the substrates. This consistency suggests uniform deposition and sensor properties across the substrate area (Supplementary Figure S14c). Additional confirmation of layer homogeneity was obtained through profilometer measurements using a Dektak device and this is presented in the Supplementary Figure S4.

Supplementary Figure S4. Homogeneity of the deposited Co layer measured by using a Dektak profilometer.

3. The authors measured the change during bending of GMR sensors to a radius of 4 mm. Considering that the sensors might experience bending repeatedly, the sensing stability was suggested to characterize under cyclic bending events to verify the device durability?

To assess the sensor durability under repeated mechanical stress, we conducted cyclic bending tests up to 2700 cycles. The GMR sensors exhibited excellent stability, with negligible change in performance after cyclic bending (Supplementary Figure S2).

The following text has been added to the manuscript.

1/ Results

The fabricated thin-film GMR sensors exhibited excellent stability, with negligible change in performance after cyclic bending (Supplementary Figure S2).

2/ In the supplementary information

Supplementary Figure S2. Thin-film GMR sensors exhibit negligible performance degradation under dynamic cyclic bending. The green shadowed area shows the resistance values during bending cycles without field and the blue shadowed area corresponds to the resistance value of the GMR sensor when a magnet (100 mT) is approached during the experiment.

4. The response time of the GMR sensor should be tested.

The frequency response of the GMR sensor has been characterized and is presented in Supplementary Figure S13, demonstrating the suitability of the sensor for real-time interactive applications.

The following text has been added to the manuscript.

1/ Results

The suitability of the sensor to operate at frequencies up to 100 kHz is demonstrated in the sensor's frequency response given in Supplementary Figure S13. This demonstrates its suitability for real-time interactive applications.

2/ In the supplementary information

Supplementary Figure S13. The change in electrical impedance of the GMR sensor as a function of frequency. The curves correspond to a sensor prepared on a rigid Si wafer and a sensor fabricated on a flexible foil. The impedance magnetic response remains stable up to 100 kHz, indicating a suitable bandwidth for real-time interactive applications.

5. The GMR sensor shows excellent performance even underwater, a practical application scenario was suggested to demonstrate to show the advantage over other sensors.

We acknowledge the value of a direct comparison. However, as most other smart textile sensors (e.g., capacitive) do not function underwater, a direct comparison through a demonstration was not possible. Moreover, attempting to test these technologies underwater without proper encapsulation could be detrimental to the sensors and even pose safety risks. Our GMR-based textiles, with their robust encapsulation, offer unique functionality in underwater environments, where conventional smart textile-based sensing technologies are unsuitable.

6. In the Introduction part, the author summarized the yarn/textile based electronics for comparison, some highly relevant references were suggested to be added for a more comprehensive background (Advanced Materials Technologies 5.1 (2020): 1900781; Advanced Functional Materials 33.44 (2023): 2303881; ACS nano 8.5 (2014): 4571-4579; Advanced materials, 2015, 27(45): 7365-7371).

We have included three of these citations in the manuscript, except for Advanced Materials Technologies 5.1 (2020): 1900781. This manuscript is on electronic skins and our focus in this paper is on smart textiles and electronic textiles.

7. The first paragraph of the Discussion part should be included into the Results part?

We agree with the reviewer. This paragraph has been moved to the results section.

8. The number of the Supplementary Figures seems out of order, please confirm it.

The manuscript is revised accordingly.

Reviewer #3

The manuscript "Submersible touchless interactivity in drapable textiles enabled by highly selective magnetosensitive smart yarns," by Lugoda and coworkers, presents a magnetoresistive sensor "integrated into yarns." According to the authors, the "Nanostructured Cu/Co giant magnetoresistive (GMR) sensors were integrated into yarns to ensure compatibility with conventional textile manufacturing equipment, continuous and reliable electrical connections, as well as to encapsulate the devices, and to preserve the feel and the aesthetics of the textile."

As a reviewer, I acknowledge the significant effort put into this manuscript. However, it currently contains numerous erroneous and misleading claims. I would recommend strongly against publication of the manuscript in its current form. Here are my specific observations;

1. From the beginning, the term 'fibertronic' is used repeatedly in the paper without offering any definition. The reviewer assumes that the authors are referring to fiber-based electronics. Later in the manuscript, it is described as a yarn in statements like 'Sensors were integrated into yarns,' 'sensor within the core of a textile yarn,' and 'magnetosensitive smart yarns,' etc. In fact, the device described by the authors is not a fiber or a yarn. What is reported is a braided tube with a sensor inside. Textiles are hierarchical structures, typically, fibers are made into yarns, yarns are processed into woven/knit fabrics, braids, and ropes, etc. The differences between these structures are significant. Describing a braided structure as a yarn is incorrect and misleading. It's crucial to use accurate terminology to avoid such misinterpretations.

We have removed the terminology like fibertronics and use the name "magnetoresistive electronic yarn" for our sensors to avoid any confusion for readers. In addition, we have added a sentence clearly defining what we mean by a magnetoresistive electronic yarn. We believe that this should avoid any confusion to the readers.

We have chosen to continue using the term "yarn" because a yarn in its simplest definition is a continuous strand comprised of natural or synthetic fibres. This is the fundamental nature of an electronic yarn, with the added component of electronics. Ultimately there are further 20-30 other journal articles using this term for braided yarns containing electronic components.

We have changed the following text in the manuscript.

1/ Abstract

Miniature electronics positioned within braided yarns leverages the persistent flexibility and comfort of textiles constructed from electronics with 1D form factors. This research enhances the functionality of textiles by developing touchless interactivity within textiles using 1D magnetoresistive electronic yarns.

2/ Introduction

Nanostructured Cu/Co giant magnetoresistive (GMR) sensors fabricated on flexible substrates were integrated within a braided textile yarns to ensure compatibility with conventional textile manufacturing equipment, continuous and reliable electrical connections, as well as to encapsulate the devices, and to preserve the feel and the aesthetics of the textile. These braided textile structures will henceforth be referred to as magnetoresistive electronic yarns.

Throughout the manuscript and supplementary information, we have referred to our device as a magnetoresistive electronic yarn.

2. What is described in this paper is a sensor made using a typical layer-by-layer electronic device fabrication (using sputtering deposition in this case) process (See page 13) with metals and polymer films. The sensors were then subsequently placed in the braided tube made of 24 polyester yarns (page 14). Lastly, the braided tubes with the sensor were inserted in pockets or “channels” of a textile garment, as explained in the sentence, “ As shown in Supplementary Figure S6, four channels were created on the knitted sleeve to incorporate the four magnetoresistive yarns.) (Page 15). It's important to accurately describe the process and components used in the study to provide a clear understanding of the research.

We have amended the manuscript to clearly indicate the process of making these magnetoresistive electronic yarns and the magnetoresistive textiles.

We have changed the following text in the manuscript.

1/ Introduction

Nanostructured Cu/Co giant magnetoresistive (GMR) sensors fabricated on flexible substrates were integrated within a braided textile yarns to ensure compatibility with conventional textile manufacturing equipment, continuous and reliable electrical connections, as well as to encapsulate the devices, and to preserve the feel and the aesthetics of the textile. These braided yarns will henceforth be referred to as magnetoresistive electronic yarns.

Multiple of these magnetoresistive electronic yarns were positioned within knitted channels of a textile sleeve to form a sensing surface, which demonstrated its capability as an interface for a virtual reality environment. Additionally, a magnetoresistive electronic yarn was sewn onto a helmet strap to showcase its potential for safety applications.

2/ Results

While magnetoresistive electronic yarns can be directly woven into the sleeve, we opted for a knitted sleeve design with integrated channels. The magnetoresistive yarns were positioned within these integrated channels. This design allowed for easier removal of the magnetoresistive electronic yarns, facilitating separate recycling of the yarns and the fabric components.

3/ A detailed explanations of the knit structure is given in Supplementary Notes 2.

3. The manuscript is full of statements that do not make sense, are erroneous, or exaggerated. Examples include

a. “The resistance values of multiple of these magnetosensitive yarns at a magnetic flux density of 14 mT is in the range of 100 Ω to 1000 Ω ” (page 7, para2). What does multiple of these mean; two, three in parallel or in series?

We have corrected this sentence. What we meant here was that we manufactured 15 yarns and their individual resistance values ranged from 100 to 1000 ohms. The following sentence has been added to the manuscript

1/ Results

The individual resistance values of 15 magnetosensitive yarns were within the range of 100 to 1000 Ω , when exposed to a magnetic field strength of 14 mT. The difference in resistance in between individual yarns is mostly due to manufacturing deviations and strain on the flexible sensors.

b. “To understand the impact of embedding sensors on the drape of the yarn and textile, a Shirley stiffness test was conducted, and the results are displayed in Figure 1e” (page 7, para 3). The test is meant to measure bending length or rigidity, it’s not a drape test.

We have removed the word ‘drape’ from the manuscript. Current drape testing methods, as outlined in established standards, are not well-suited for evaluating the drape of E-textiles (as mentioned in <https://doi.org/10.3390/textiles4020013>). As a result, we opted for a bending rigidity test, which offered some insights into drape behavior, even though it’s not a direct measurement.

We have added the following text to the manuscript. We have removed the word ‘drape’ and ‘drapable’ from the manuscript.

1/ Results

However, the current drape testing methods and established standards, are not well-suited for evaluating the drape of E-textiles.^[66] As a result, we opted for a bending rigidity test, which offered some insights into drape behaviour, even though it's not a direct measurement. To understand the impact of embedding sensors on the bending rigidity of the yarn and textile, a Shirley stiffness test was conducted, and the results are displayed in Figure 1e.^[67]

c. “The data indicate that the bending rigidity of the yarn increased by 23% after the integration of the magnetoresistive sensor (page 7, last para).” The statement is problematic at many levels. The 23% increase in bending rigidity from that of the braided tube? How does that matter in terms of wearability? The rigidity of the braided tube with the sensor in it is very likely to vary along its length. What did the authors consider.” The last sentence was followed by the statement, “Nonetheless, once the yarns were inserted into a fabric, the bending rigidity was measured at (3155 ± 437) mg cm along the vertical axis and (1670 ± 160) mg cm along the horizontal axis of the textile sleeve. This level of rigidity falls below that of certain woven structures crafted solely from traditional textile materials.[63] This shows that our magnetoresistive textiles are highly conformable and therefore comfortable to wear (page 7, last para).” Most of the fabrics reported in reference 63 (Sule et al.) are less rigid than those reported in this paper. The only group of fabrics reported to have higher bending rigidity is an outlier made of really heavy yarns with high-end/pick density. It is entirely misleading to claim that “magnetoresistive textiles are highly conformable and therefore comfortable to wear” based on this information.

Following these comments of the reviewer, we have changed the following sentences.

1/ Results

This level of rigidity falls below that of certain woven structures crafted from heavy yarns.[68] While these magnetoresistive textiles may not be as flexible as traditional fabrics, they offer a reasonable level of conformability, ensuring that they don't significantly compromise wearer comfort.

Supplementary Figure S11 displays images of the textile sleeve containing the magnetoresistive electronic yarns conforming around a mannequin forearm.

2/ In the supplementary information

Supplementary Figure S11 demonstrates the textile sleeve containing the magnetoresistive electronic yarns conforming around a mannequin forearm.

Supplementary Figure S11: Display images of the textile sleeve containing four magnetoresistive electronic yarns worn on a mannequin hand. The sleeve fits snugly and conforms to the shape of a mannequin hand.

d. “The scaling of the production of several meters of yarn can be realized through high throughput fabrication of magnetic field sensors over large area flexible substrates of up to 300 mm (Supplementary Figure S7).” How does 300 mm translate to several meters?

In microelectronics, a 300 mm wafer substrate is a standard substrate size used for high-throughput production. For example, with a sensor footprint of 1 cm², a 300 mm wafer, which has an area of about 70,650 mm², can accommodate roughly 7,000 sensors. Assuming one sensor is placed per centimeter of yarn, this allows for the production of approximately 70 meters of yarn with embedded sensors from a single 300 mm wafer.

We have changed the following text in the manuscript.

1/ Results

Scaling up the production of several meters of yarn can be achieved by fabricating magnetic field sensors on large-area flexible substrates, up to 300 mm in diameter (see Supplementary Figure S14). With a sensor footprint of 1 cm², a 300 mm wafer, which has an area of about 70,650 mm², can accommodate roughly 7,000 sensors. Assuming one sensor is placed per centimeter of yarn, this allows for the production of approximately 70 meters of yarn from a single substrate wafer.

Response letter

We thank the reviewers for their positive assessment of the revised version of the manuscript. We appreciate the recommendation of the Reviewer #2 to publish our manuscript. Furthermore, we thank the Reviewer #3 for his/her constructive input regarding the terminology. Following the suggestion of the Reviewer #3, we changed wording and instead of using “yarn” we use the expression “overbraided sensor”. The manuscript is revised accordingly. With these changes we hope that the manuscript can be recommended for publication.

Comments on manuscript COMMSENG-24-0332-T

The manuscript "Submersible touchless interactivity in drapable textiles enabled by highly selective magnetosensitive smart yarns," by *Lugoda* and coworkers, presents a magnetoresistive sensor "integrated into yarns." According to the authors, the "Nanostructured Cu/Co giant magnetoresistive (GMR) sensors were integrated into yarns to ensure compatibility with conventional textile manufacturing equipment, continuous and reliable electrical connections, as well as to encapsulate the devices, and to preserve the feel and the aesthetics of the textile."

As a reviewer, I acknowledge the significant effort put into this manuscript. However, it currently contains numerous erroneous and misleading claims. I would recommend strongly against publication of the manuscript in its current form. Here are my specific observations;

1. From the beginning, the term 'fibertronic' is used repeatedly in the paper without offering any definition. The reviewer assumes that the authors are referring to fiber-based electronics. Later in the manuscript, it is described as a yarn in statements like 'Sensors were integrated into yarns,' 'sensor within the core of a textile yarn,' and 'magnetosensitive smart yarns,' etc. In fact, the device described by the authors is not a fiber or a yarn. What is reported is a braided tube with a sensor inside. Textiles are hierarchical structures, typically, fibers are made into yarns, yarns are processed into woven/knit fabrics, braids, and ropes, etc. The differences between these structures are significant. Describing a braided structure as a yarn is incorrect and misleading. It's crucial to use accurate terminology to avoid such misinterpretations.
2. What is described in this paper is a sensor made using a typical layer-by-layer electronic device fabrication (using sputtering deposition in this case) process (See page 13) with metals and polymer films. The sensors were then subsequently placed in the braided tube made of 24 polyester yarns (page 14). Lastly, the braided tubes with the sensor were inserted in pockets or "channels" of a textile garment, as explained in the sentence, "As shown in Supplementary Figure S6, four channels were created on the knitted sleeve to incorporate the four magnetoresistive yarns.) (Page 15). It's important to accurately describe the process and components used in the study to provide a clear understanding of the research.
3. The manuscript is full of statements that do not make sense, are erroneous, or exaggerated. Examples include
 - a. "The resistance values of multiple of these magnetosensitive yarns at a magnetic flux density of 14 mT is in the range of 100 Ω to 1000 Ω " (page 7, para2). What does multiple of these mean; two, three in parallel or in series?
 - b. "To understand the impact of embedding sensors on the drape of the yarn and textile, a Shirley stiffness test was conducted, and the results are displayed in Figure 1e" (page 7, para 3). The test is meant to measure bending length or rigidity, it's not a drape test.
 - c. "The data indicate that the bending rigidity of the yarn increased by 23% after the integration of the magnetoresistive sensor (page 7, last para)." The statement is problematic at many levels. The 23% increase in bending rigidity from that of the braided

tube? How does that matter in terms of wearability? The rigidity of the braided tube with the sensor in it is very likely to vary along its length. What did the authors consider.” The last sentence was followed by the statement, “Nonetheless, once the yarns were inserted into a fabric, the bending rigidity was measured at (3155 ± 437) mg cm along the vertical axis and (1670 ± 160) mg cm along the horizontal axis of the textile sleeve. This level of rigidity falls below that of certain woven structures crafted solely from traditional textile materials.[63] This shows that our magnetoresistive textiles are highly conformable and therefore comfortable to wear (page 7, last para).” Most of the fabrics reported in reference 63 (Sule et al.) are less rigid than those reported in this paper. The only group of fabrics reported to have higher bending rigidity is an outlier made of really heavy yarns with high-end/pick density. It is entirely misleading to claim that “magnetoresistive textiles are highly conformable and therefore comfortable to wear” based on this information.

- d. “The scaling of the production of several meters of yarn can be realized through high throughput fabrication of magnetic field sensors over large area flexible substrates of up to 300 mm (Supplementary Figure S7).” How does 300 mm translate to several meters?

There are many more of these examples. For the sake of time, I must stop here.

Comments on manuscript COMMS-24-0332-A

The revised manuscript, with a revised title, "Submersible touchless interactivity in conformable textiles enabled by highly selective magnetoresistive electronic yarns," contains most of the changes suggested by the reviewers. Once again, I acknowledge the significant efforts behind the sensor concept presented in this manuscript. However, I would again request the authors to describe their work for the reader's benefit appropriately. Despite the reviewer's earlier suggestion to appropriately describe the braided textile structure as a braid rather than a yarn, the authors decided to continue using the term yarn to describe a braid made of yarn. The description is erroneous and misleading and, therefore, must be changed. I would strongly recommend against the publication of the manuscript in its current form.

What is described now as a "magnetoresistive electronic yarn," is actually a braided tube with a sensor inside. **Textiles are hierarchical structures; typically, fibers are made into yarns, and yarns are processed into woven/knit fabrics, braids, ropes, etc.** The distinction between these structures is a topic for Textiles 101. The authors justify their continuing use of the term yarn "because a yarn in its simplest definition is a continuous strand comprised of natural or synthetic fibres." Obviously, using this definition, a rope is a yarn. The other justification presented by the authors is "there are further 20-30 other journal articles using this term for braided yarns containing electronic components." I am not sure where those 20-30 journal articles have been published. Although, in today's publishing environment, it is plausible that an error by one gets repeated by many. However, mistakes by many do not make it right. **There are NO braided yarns; braids are made of yarns.**

Here is how a yarn and a braid is defined and distinguished in many publications and standards;

"In the textile industry yarn is an assemblage of natural or man-made fibers or filaments that are twisted together to form a continuous strand, which can be used in knitting, weaving, braiding or pleating or can be made into a textile in some other way."

Stanley Baker, YARN, Scientific American, Vol. 227, No. 6 (December 1972), pp. 46-57, Published by: Scientific American, a division of Nature America, Inc.

"A textile yarn may be defined as a long fine fiber strand of parallel continuous filaments or twisted staple fibers. These linear structures can be interlaced or intermeshed into a two-dimensional or three-dimensional woven or knit fabric structures, and they can also be intertwisted into braids, ropes, or cords."

Yarns, Yehia E. Elmogahzy, in Engineering Textiles, Integrating the Design and Manufacture of Textile Products, A volume in The Textile Institute Book Series, Woodhead Publishing, DOI <https://doi.org/10.1016/C2017-0-01393-9>

"a generic term for a continuous strand of textile fibers, filaments, or material in a form suitable for knitting, weaving, or otherwise intertwining to form a textile fabric."

ASTM D4849-21, Standard Terminology Related to Yarns and Fibers

"A textile yarn is a continuous strand of staple or filament fibers arranged in a form suitable for weaving, knitting, or other form of fabric assembly. Also, a yarn is a textile product of substantial length and relatively small cross-section consisting of fibers with twist and/or filaments without twist. The yarn can

be twisted with one or more yarns to create added value or aesthetics. Traditionally, yarns have been constructed of fibers of finite length called staple fibers. Today, continuous filament yarns are also used to construct yarns."

Textile Yarns 101 (form Cotton Incorporated)

"Braid is a minor but distinctive form of textile cloth."

"A narrow tubular or flat fabric produced by intertwining a single set of yarns according to a definite pattern."

"A structure produced by interlacing several ends of yarn in a manner such that the paths of the yarns are not parallel to the braid axis."

D. Brunnschweiler (1953) BRAIDS AND BRAIDING, Journal of the Textile Institute Proceedings, 44:9, P666-P686, DOI: 10.1080/19447015308687874

*"Braiding is a process of interlacing three or more threads diagonally to the product axis (parallel to the longest dimension of the resulting product) in order to obtain a thicker, wider, or stronger product or in order to cover (**overbraid**) some profile."*

Braiding Technology for Textiles, Y. Kyosev, Woodhead Publishing in association with The Textile Institute, Cambridge, CB22 3HJ, UK

Overbraiding is a commonly used term to describe, covering a core material with a braided structure, including cable-sleeving, and electrical wire, etc. This is exactly what the authors have done in this work. An appropriate term to describe their proposed structure is an overbraided sensor